# Natural Products in Mitigation of Bisphenol A Toxicity: Future Therapeutic Use

**DOI:** 10.3390/molecules27175384

**Published:** 2022-08-24

**Authors:** Srinivasa Rao Sirasanagandla, Isehaq Al-Huseini, Hussein Sakr, Marzie Moqadass, Srijit Das, Norsham Juliana, Izuddin Fahmy Abu

**Affiliations:** 1Department of Human and Clinical Anatomy, College of Medicine and Health Sciences, Sultan Qaboos University, Muscat 123, Oman; 2College of Medicine and Health Sciences, Sultan Qaboos University, Muscat 123, Oman; 3Faculty of Medicine and Health Sciences, Universiti Sains Islam Malaysia, Nilai 71800, Malaysia; 4Institute of Medical Science Technology, Universiti Kuala Lumpur, Kuala Lumpur 50250, Malaysia

**Keywords:** bisphenol A, pollutant, toxicity, treatment, natural product, plastics

## Abstract

Bisphenol A (BPA) is a ubiquitous environmental toxin with deleterious endocrine-disrupting effects. It is widely used in producing epoxy resins, polycarbonate plastics, and polyvinyl chloride plastics. Human beings are regularly exposed to BPA through inhalation, ingestion, and topical absorption routes. The prevalence of BPA exposure has considerably increased over the past decades. Previous research studies have found a plethora of evidence of BPA’s harmful effects. Interestingly, even at a lower concentration, this industrial product was found to be harmful at cellular and tissue levels, affecting various body functions. A noble and possible treatment could be made plausible by using natural products (NPs). In this review, we highlight existing experimental evidence of NPs against BPA exposure-induced adverse effects, which involve the body’s reproductive, neurological, hepatic, renal, cardiovascular, and endocrine systems. The review also focuses on the targeted signaling pathways of NPs involved in BPA-induced toxicity. Although potential molecular mechanisms underlying BPA-induced toxicity have been investigated, there is currently no specific targeted treatment for BPA-induced toxicity. Hence, natural products could be considered for future therapeutic use against adverse and harmful effects of BPA exposure.

## 1. Introduction

### 1.1. Bisphenol A

Bisphenol A (BPA) and its derivatives, bisphenol S, bisphenol AF, bisphenol E, and bisphenol B, consist of two phenyl rings connected by a small linking group. This synthetic carbon-based compound is widely used in various industries [1]. It was reported that the production of this compound was estimated to be 4.85 million tons [2]. Since the 1960s, BPA has been used in epoxy resins and polycarbonate plastics production, which are the main constituents of daily consumer products, including containers of food products and drinks, thermal receipts, medical equipment, toys, dental sealants, paints, CD/DVDs, etc. [3,4]. BPA biodegrades, leaches easily from consumer goods, and enters the environment. Its presence has been detected in soil, both drinking and wastewater, dust, food, and air [5].

BPA has a half-life of approximately 4.5 days in water and soil and less than one day in the air [6]. At room temperature, leaching of BPA from polycarbonate bottles into drinking water occurred at an observed rate of 0.20 to 0.79 ng/h. At boiling temperature, the leaching of BPA from the same polycarbonate bottles into water increases 55-fold [7]. Researchers also found that, along with temperature, repeated use of a bottle will influence the leaching of BPA [8]. BPA can also enter food stuff through the BPA epoxy resin films, which are used in the lining of food cans [9,10,11]. Other BPA exposure sources include thermal printing paper [12], medical equipment, and plastics used in medical devices [13], particularly polycarbonate equipment used in hemodialysis [14], and dental composites and sealants [15]. According to the European Food Safety Authority (EFSA), the daily tolerated intake of BPA was reported to be 4 μg/kg/day [16].

BPA may be exposed through several exposure routes, including the integumentary system, digestive system, respiratory system, and maternofetal transmission [17]. It is easily absorbed in the gastrointestinal tract, whereby it is metabolized and secreted within a few hours of exposure. In the liver, BPA is biotransformed into BPA-glucuronide after conjugating with the uridine diphosphate glucuronic acid. BPA is also known to form other conjugates, such as BPA diglucuronide and BPA sulfate conjugates. The conjugate formation and excretion of bisphenol glucuronide are rapid and take place with a half-life of 5.3 h [18,19]. BPA conjugates are excreted in the bile, urine, and feces. The developing fetus, as well as infants, are also exposed to BPA through placental transfer and milk, respectively [20]. The β-glucuronidase enzyme converts conjugated BPA into its active free form. A considerable amount of this enzyme has been detected in various tissues, including rodent livers, lungs, kidneys, and human and rodent placentas [21]. This could be one of the reasons that even though BPA is rapidly eliminated in more than 95% of individuals, considerable levels of BPA can still be observed [4,22]. Consequently, human fluids and tissues only contain very little non-conjugated BPA, often in the nanogram per milliliter range [23,24]. According to a study, the mean free BPA concentration in human serum ranged from 2.3 to 2.4 ng mL^−1^, 2.8 ng mL^−1^ in adolescents, and 4.3 ng mL^−1^ in children [25]. A BPA biomonitoring study has shown that the compound can be found in human blood, urine, and milk in nanomolar quantities [24]. Of great concern is that BPA could be found in various bodily fluids from vulnerable populations, including in the urine of children and healthy infants, pregnant women’s serum, fetal serum, follicular and amniotic fluid, cord blood, placenta tissue, breast milk, and saliva from patients who have undergone certain dental procedures [26]. BPA was found to accumulate more in fat compared to other tissues, such as kidneys, muscles, and others due to its high affinity with the fat [27]. In a study conducted by Geens et al., free BPA levels were detected highest in adipose tissues (1.12 to 12.28 ng/g), followed by the liver (0.77 to 3.35 ng/g), and brain (up to 2.36 ng/g). In breast milk, total BPA and unconjugated BPA were detected at concentrations of 1.1 and 0.4 ng/mL, respectively [28]. A study undertaken to determine the distribution of BPA in people revealed that BPA was found in almost all human tissues. Due to its tendency to bioaccumulate in human tissues, BPA can potentially pose long-term metabolic consequences.

Ever since the 1990s, BPA occurrence has been detected in the environment and continues to contaminate and pollute globally [29]. As a primary river pollutant, BPA’s presence has been observed in surface water and drinking water. A study from Malaysia reported a presence of BPA in the Bentong River at concentrations of 5.52 and 2.06 ng/L in colloidal and soluble phases, respectively [30]. In studies from India, Spain, Germany, and China, the BPA levels in the surface water were reported to be in the range of 6.63–136 ng/L, 87–126 ng/L, 28–560 ng/L, and 22–3360 ng/L, respectively [31,32,33,34]. In a recent study, in two different rivers in Romania, the BPA concentrations were found to be in the range of 22.1–35.8 ng/L and 74.5–135 ng/L, respectively [35]. BPA can reach the soil through sewage sludge from wastewater treatment plants, leachate from landfills, and soil amendments [36]. Studies from Asian countries, such as Korea, China, Japan, and India have reported BPA levels of 0.5–48.68 μg/kg, 2340 pg/m^3^, 1920 pg/m^3^, and 17,400 pg/m^3^, respectively [36]. In Mexico, BPA concentrations in agricultural soil irrigated with wastewater were found to be 1.6–30.2 μg/kg [37]. In European soils that were amended with biosolids, a median concentration of 0.24 μg/kg BPA was reported [38]. BPA can enter the air, and most of its sources are anthropogenic. The contamination of BPA into the atmosphere occurs through various routes including incineration of plastic materials, transportation of recyclable materials and environmental emissions, BPA-producing factories, waste treatment plants, and landfills [29,39]. The highest air levels of BPA were documented in China at a concentration of 1.1 × 10^6^ pg/m^3^ [40].

### 1.2. Conventional Methods for BPA Degradation and Removal

BPA is potentially harmful at concentrations ranging from 5 to 200 n/L (nanoliters) in surface and ground waters [41]. BPA may be accumulated in aquatic organisms. Since BPA may accumulate in the water, better water treatment methods are essential. Biodegradation has several advantages, such as environment and economic protection, low costs, wider scopes of action, longer durations, and fewer problems with regard to space and equipment requirements [42]. White-rot fungi can produce extracellular lignin-producing enzymes, which are responsible for the biotransformation of many aromatic compounds and pollutants, such as BPA [43,44]. It was found that 80% of BPA was removed over a period of 12 days using white-rot basidiomycete Pleurotus ostreatus [45]. Many enzymes, such as lignin peroxidase (LiP), manganese-dependent peroxidase (MnP), and laccase may be beneficial in the degradation of BPA. MnP enzyme was reported to catalyze the oxidation of various phenols in the presence of H_2_O_2_ and Mn(II) [46]. MnP is a heme peroxidase enzyme and oxidizes phenolic compounds in the presence of Mn(II) and H_2_O_2_, while laccase is a multicopper oxidase enzyme and catalyzes one-electron oxidation of phenolic compounds by reducing oxygen to water [47]. Compared to MnP, laccase remains stable and high throughout the incubation period, suggesting that it may be a key enzyme in BPA degradation [48]. In the future, probiotics could be tried for BPA degradation.

Fly ash represents 80% of the coal combustion byproducts. Besides its utilization for building and construction purposes, fly ash has beneficial effects on environmental and economic applications [49]. Interestingly, fly ash has been used to degrade and remove phenol chemicals, such as the endocrine disrupter BPA [50]. A zeolite is prepared from fly ash and has a low absorption capacity of BPA. However, the absorption capacity of BPA by zeolites was greatly augmented after the surfactant modification [51]. Moreover, the absorption of peroxidase on fly ash improves its effectiveness in degrading BPA when compared to its free form [52]. Fly ash is highly effective and stable for the oxidative polymerization and removal of BPA and other toxicants.

## 2. Effects of BPA Exposure on Different Body Systems

### 2.1. The Changes Occuring in Different Systems of the Body

#### 2.1.1. Reproductive System

The roles of BPA in endocrine disruption, oxidative stress, epigenetic modification, the release of cytokines, and oxidative stress are linked to its associated adverse effects. BPA is well known for its estrogenic activity. BPA was initially explored for its estrogenic properties and was found to influence the synthesis of estrogen and testosterone [53,54]. In subsequent studies, BPA was found to interfere with sex hormone activities and associated with developmental toxicity and functional disturbances of the reproductive system. BPA is responsible for the pathogenesis of female infertility [55]. In a study, serum BPA levels were detected (limit of assay detection: 0.5 ng/mL) in 41.8% of infertile women and 23.3% of fertile women [56]. Interestingly, higher BPA exposure and levels in infertile women in a metropolitan area show evidence of a greater presence of economic activities using these chemicals and more usage in food and consumer products [56]. It cannot be refuted that BPA may be detected in infertile women. In females, the BPA-induced reproductive abnormalities include increased endometrial wall thickness, the occurrence of polycystic ovary syndrome, an increased risk of recurrent miscarriage, neonatal mortality, defective placental function, irregular cycles, and reduced primordial follicles [57,58,59,60,61,62,63,64]. Experimental and epidemiological studies have confirmed that BPA exposure during pregnancy affects the development and growth of offspring. Exposure to BPA in utero has been shown to affect the development of the uterus and mammary glands [65,66]. A direct association was also found between urinary BPA levels and implantation failure [67]. In males, BPA can interfere with the regulation of spermatogenesis via the hypothalamic–pituitary–gonadal axis. BPA has been shown to impair male reproductive function with a reduction in sperm quality, defective ejaculation, reduced libido, and erectile dysfunction [68,69]. In experimental animals, administration of BPA significantly decreased the expression of the GnRH gene in cells of the preoptic area and circulating levels of gonadotropins and/or testosterone [70]. Occupational exposure to BPA among adult males in China has been reported to be associated with changes in serum hormone levels and male sexual dysfunction [69].

#### 2.1.2. Cardiovascular System

Prenatal BPA exposure was also linked to an increased risk of developing cardiovascular disorders and non-alcoholic liver disease in later life [71,72,73,74,75]. Furthermore, BPA-induced neuroendocrine regulation may result in mental and behavioral consequences in the offspring. Higher maternal BPA levels increase the risk of developing behavior problems in preschool children [76]. Prenatal BPA exposure was also strongly associated with anxiety and depression in children [77]. Clinical evidence shows an association between serum and/or urinary BPA levels and cardiovascular diseases [78,79]. Increased serum BPA levels in dilated cardiomyopathic patients were also reported [80]. Experimental studies confirmed that BPA exposure could adversely affect cardiac structure and function [81,82]. Furthermore, BPA exposure increases systolic blood pressure, alters heartbeat in isolated heart preparations, and blocks cardiac sodium channel receptors [83,84,85]. BPA can also alter calcium homeostasis in the heart by stimulating estrogen receptors on the plasma membrane [86]. BPA exposure was also associated with the development of atherosclerosis and peripheral arterial disease [87,88,89].

#### 2.1.3. Endocrine System

Recent animal studies have observed that BPA can cause developmental programmings of metabolic diseases, such as diabetes mellitus and obesity in later stages of life [74,90,91]. BPA exposure affects glucose metabolism by disrupting pancreatic cell function and producing insulin secretion [92]. It also affects adipocytes’ metabolic functions, leading to insulin resistance development. In an in vivo study, BPA exposure promoted insulin resistance by reducing adiponectin levels and increasing adipocytokines levels [93]. Urinary BPA levels were positively linked with metabolic syndrome [88]. Few epidemiological studies have found a link between BPA exposure and diabetes mellitus in patients who were not predisposed to the disease due to factors, such as age, serum cholesterol levels, or body mass index [88,94,95]. In obese pregnant women, BPA exposure was associated with an increased risk of altered glucose metabolism [96]. Furthermore, it was found that BPA shows these effects in specific trimesters, particularly in the second trimester [97]. Higher urinary BPA levels were associated with childhood obesity as well as abdominal obesity [98]. In addition, BPA perinatal exposure induces the development of diabetes mellitus in the offspring. BPA-induced developmental programming is thought to be due to epigenetic modifications [99].

#### 2.1.4. Urinary System

Since BPA is a xenoestrogen and the kidneys possess several receptors for estrogen, BPA stimulates the receptors, leading to the proliferation of epithelial cells and also increases the volume of proximal and distal cells leading to hydronephrosis [100]. BPA treatment in animals showed bladder enlargement, hypertrophy, and urinary voiding dysfunction [101]. The late manifestations of voiding dysfunction were also observed in mice treated with testosterone and BPA [101]. The authors observed that in adulthood, BPA exposure is associated with lower urinary tract dysfunction [101]. Interestingly, narrowing of the lumen of the prostatic urethra of mice treated with BPA testosterone was observed [101]. This may be similar to prostate enlargement and bladder alterations in humans. Enlargement of the prostate and urinary bladder may decrease the average flow rate. Higher BPA levels were strongly associated with reduced glomerular filtration rate and impaired renal function [102]. Chronic BPA exposure caused inflammatory infiltration, fibrosis, and tubular injury in the kidney. BPA-induced defective autophagy flux was the key mechanism behind these effects [103].

#### 2.1.5. Gastrointestinal System

In vitro studies have shown that BPA promotes mitochondrial dysfunction, oxidative stress, and inflammation [104,105]. BPA is well known to affect liver enzymes’ activities and promote hepatic lipid accumulation [106]. High urinary BPA levels were found to be associated with non-alcoholic fatty liver disease in adults [107]. BPA has been demonstrated to promote oxidative phosphorylation abnormalities in the liver mitochondria by inhibiting the first complex of the electron transport chain [108]. In several studies, BPA has been shown to pose deleterious effects on liver function and structure in both people and animals. BPA can promote hepatic steatosis in humans [109], enhance insulin resistance in HepG2 cells [110], alter its shape, and raise liver function enzymes [111]. The upregulation of sterol regulatory element binding protein 1 has been implicated in BPA-induced hepatic lipid accumulation [112].

#### 2.1.6. Immune System

As polymorphonuclear neutrophils (PMN) are essential in stimulating the congenital immune response, various studies looked into the effects of BPA on PMN. Findings of one study showed that BPA exposure (of more than 16 μM) caused a decline in human polymorphonuclear neutrophils (PMN) viability and demonstrated morphological alterations in these cells in both sexes [113]. A subsequent study revealed that BPA alters the immunophenotype of PMN at a dose corresponding to the serum level in healthy subjects and also at higher doses, which may eventually cause immunity problems linked to the malfunctioning of these cells [114]. BPA has also been demonstrated to exert a direct genotoxic effect in human lymphocytes by inducing the double-strand breaks of the DNA [26]. Another in vitro study revealed a significant increase in reactive oxygen species (ROS) production in erythrocytes when BPA concentration exposed was increased from 1 to 100 μM [115].

BPA exposure is known to exert complex modulatory effects on the immune system, with stimulatory or suppressive roles [116]. BPA exposure was positively linked with increased serum IgE levels [117]. In addition, higher urinary BPA levels were associated with childhood asthma [118]. Perinatal BPA exposure has been shown to increase the development of asthma and allergic disorders in children [119,120]. Studies have also shown that BPA can increase the risk of developing cancers, including prostate, breast, ovarian, lung, cervical cancer, etc. [5,121,122]. The estrogenic activities of BPA have been implicated in the mechanism behind BPA-induced cancers. In another study, BPA exposure was positively associated with the risk of developing an autism spectrum disorder (ASD) [123]. In addition, BPA was observed to increase oxidative stress-induced mitochondrial dysfunction, leading to behavioral changes in children with ASD [124].

#### 2.1.7. Respiratory System

It has been found that postnatal exposure to BPA is a risk factor for the development of childhood asthma [125]. BPA has been reported to be associated with bronchial eosinophilic inflammation/allergic sensitization [126]. A research report published in 2019 showed that a doubling of BPA in a mother’s urine sample corresponded with (approximately) a 5 mL decrease in the child’s lung capacity [127]. In adult murine asthma models, researchers showed an aggravating effect of BPA on eosinophil infiltration and airway inflammation as evidenced by increasing levels of Th_2_ cytokines and chemokines [128]. The same study found that BPA affected allergic inflammation in allergic asthmatics [128]. Hence, BPA exposure may have detrimental effects on the respiratory system.

#### 2.1.8. Nervous System

Recent research studies found that a derivative of BPA, i.e., Bisphenol F, is responsible for neuroinflammation and apoptosis of central nervous system cells, leading to abnormal neurological development in the early life stage of zebrafish [129]. A disturbance in the neurotransmitter function may be responsible for the disturbance in the nervous system. One of the neurotransmitters, GABA, is responsible for maintaining the balance between the excitatory and inhibitory systems necessary for the development of a normal brain [129]. Research studies on animal models have shown that prenatal exposure to BPA affects the mevalonate (MVA) pathway in rat brain fetuses [130]. The MVA pathway is important for the development and function of the brain [130]. Interestingly, hypothalamic exposure to BPA showed an increase in micro RNA (miRNA) miR-708-5p, which is responsible for controlling neuropeptides directly linked to obesity [131].

### 2.2. Underlying Mechanisms of BPA Exposure-Induced Toxicity

In 1936, Dowds and Lawson were the first to describe BPA’s estrogenic properties in vivo [132]. Later in 1997, BPA was found to act through estrogen receptors, Erα, and ERβ [133,134]. BPA is considered a “weak estrogen” as it exerts a weaker affinity with the estrogen receptors when compared to estradiol. Several investigations have been conducted into other possible mechanisms. Alternatively, BPA binds to the membrane estrogen receptor (mER) and activates non-ERS-dependent signaling pathways, causing biological dysfunction even at picomolar to nanomolar concentrations [135]. These concentrations are lower than those necessary to stimulate nuclear ERs [136]. The G protein-coupled estrogen receptor is a mER that plays a major role in BPA-induced toxicity [137].

These studies have shown that BPA-induced adverse health effects are mediated through its potential binding capacity with various nuclear receptors, including estrogen-related receptor (ERRγ), androgen receptor (AR), thyroid receptors (TRα and TRβ), glucocorticoid receptor (GR), and mineralocorticoid receptor (MR) [133,138,139]. The binding capacity of BPA with ERRγ is 80-fold higher than ERα [140]. This explains the high BPA levels in the placenta and its placental transfer and subsequent developmental effects [141]. BPA binds with AR to form an AR/BPA complex. This complex inhibits the endogenous androgen-mediated gene transcription, which is the key mechanism behind the BPA-induced anti-androgenetic effects [142,143]. BPA can bind to TR, and by acting as an antagonist, inhibit the TR-mediated transcription [144]. Similar to cortisol and dexamethasone, BPA can bind to GR and promote glucocorticoid-mediated biological functions [145]. BPA has been shown to alter the GR-linked feedback of the hypothalamic–pituitary–adrenal axis through GR expression regulation. BPA was also found to alter the expression of MR [146].

The genotoxic mechanisms of BPA are well established in the literature [147,148,149]. BPA has been shown to induce oxidative damage in human lung fibroblasts and promote DNA damage in human epithelial type 2 cells [149]. In human hepatocytes, BPA exposure at a very low concentration induced DNA damage as well as increased proliferation by enhancing cell-cycle protein expression and DNA synthesis [148]. BPA is involved in the dysfunction of certain enzymes. BPA promotes oxidative damage by reducing antioxidant enzymes [150]. BPA has been shown to influence the function of xanthine oxidase [151], lipoprotein lipase (Lpl), fatty acid acetyl-coenzyme A carboxylase β synthase [152], and fatty acid amide hydrolase [153]. BPA can cause three forms of epigenetic modifications: DNA methylation, histone modification, and miRNA alterations [154]. Growing evidence suggests that BPA-induced autophagy modulation is associated with the pathogenesis of some diseases. However, the molecular mechanisms involving BPA-induced autophagy modulation are yet to be fully understood. This may help in future treatments and drug discoveries [155,156].

## 3. Various Natural Products That Are Effective against BPA-Induced Toxicity

Evidence from molecular studies indicates that BPA affects various signaling pathways that are closely associated with the pathogenesis of chronic diseases. Therefore, it is important to develop new strategies that modulate/inhibit the BPA-induced toxic effects on the pathophysiologic processes of a disease. In this line of research, various studies have been performed to explore the effects of natural products against the BPA-induced toxic effects using different experimental models, including rodents, Drosophila, and zebrafish [Table 1]. In this section, we describe the effects of plant extracts or natural products against BPA-induced toxicity in two subsections: the ameliorative potential of plant extracts and natural compounds.

### 3.1. Plant Extract/Mixture of Natural Compounds

#### 3.1.1. *Pistacia integerrima*

*Pistacia integerrima* is a small size tree in the family of cashew Anacardiaceae, grown in the northern area of Pakistan [157]. It has various biological functions, including antioxidant, analgesic, anti-inflammatory, and anti-microbial activities [158]. Traditionally, some of its parts have been used to treat different diseases, such as liver disorders, asthma, snake bites, and cough [159]. Despite the large number of reports on the biological activities of *P. integerrima*, only one study has been conducted to explore its biological activity on BPA toxicity. It was demonstrated that *P. integerrima* ameliorates BPA exposure-induced cardiotoxicity in rats by neutralizing oxidative stress and suppressing apoptosis through the Ubc13/p53 pathway in rats [157]. The anti-apoptotic effects of *P. integerrima* are shown in Figure 1.

#### 3.1.2. Fenugreek (*Trigonella foenum-graecum*)

Fenugreek (*Trigonella foenum-graecum*) belongs to the Fabaceae family. It is a short-lived annual plant. It is one of the most well-known medicinal plants known for its healing benefits [161]. To date, several studies have focused on the health benefits of fenugreek. Previous studies have reported the anti-inflammatory [162], antioxidant [163], antidiabetic [164], anticancer [165], anti-obesity [166], hepatoprotective [167], women’s health [168], anti-hyperlipidemic [169], and sexual health-modulating activities [170] of the plant. Fenugreek seed extract treatment in mice prevented BPA exposure-induced testicular damage by decreasing malondialdehyde levels and increasing the levels of antioxidant enzymes [171].

#### 3.1.3. Kefir

Kefir is a probiotic drink known for its health benefits. Kefir is produced through the fermentation of milk with kefir grains. Kefir grains contain various bacterial and fungal species. Kefir is known to reduce inflammation and serum cholesterol levels. It has anti-carcinogenic effects and can enhance gut health and digestion. It can also reduce hypertension and regulate reactive oxygen species. Lactobacillus species are the main bacterial flora in kefir [172]. The benefits of kefir are attributed to the complex microbiota and metabolites produced during the fermentation process [173]. In infant rats, probiotic kefir treatment attenuated the progression of BPA exposure-induced hypertension and vascular changes, including vascular ROS/NO imbalance, endothelial dysfunction and damage, and pro-apoptotic effects [174].

#### 3.1.4. Grape Seed (*Vitis vinifera* L.)

Grape seed proanthocyanidins (GSP) include a complex combination of polyphenolic compounds [175]. GSP is known for its protective effects against lipid peroxidation and DNA damage, particularly in the brain and liver. In addition, it has antibacterial, antidiabetic, anti-inflammatory, and anticancer effects [176]. GSP exerts protective effects against brain damage through its antioxidant properties and chelating ability [177]. GSP has a protective role against depression and age-related mental disorders by promoting hippocampal neurogenesis [178]. A study has shown that BPA exposure caused neuroinflammation, neuronal tissue damage, increased oxidative stress, and altered the levels of neuro-specific enzymes and neurotransmitters. Treatment with GSP significantly prevented BPA-induced neurotoxicity by ameliorating all oxidative and neurotoxic parameters in Wistar rats [179].

Grape seed extracts (GSEs), which are high in flavonoids, particularly proanthocyanidin, have been demonstrated to possess potent antioxidant properties [180]. Oral treatment of GSE reduces ROS production and plasma protein carbonyl groups while increasing the activity of the endogenous antioxidant system [180]. GSE has antimutagenic and anticarcinogenic properties because it inhibits enzyme systems that produce free radicals. It protects against oxidant-induced extracellular matrix component synthesis and deposition, leading to hepatic fibrosis. GSE’s antioxidant properties have been established in clinical investigations [181]. GSE has been shown to possess anti-inflammatory and antioxidant properties, which may explain its therapeutic efficacy in a collagen-induced arthritis animal model [181]. In an isolated rat model, BPA exposure induced vascular toxicity by decreasing the vasoconstriction and vasorelaxation responses and elevating the aorta MDA levels, as well as via an in vitro analysis—BPA induced endothelial dysfunction by increasing the adhesion molecules levels [182]. In rats, RSV and GSE significantly prevented vascular toxicity [182] and metabolic syndrome [183]. The beneficial effects of GSE and RSV against BPA-induced metabolic syndrome were associated with antioxidant properties, insulin signaling regulation, and *ABCG*8 expression [183]. The schematic representation of insulin signaling promoting the potential of resveratrol and GSE is shown in Figure 2.

#### 3.1.5. *Ficus deltoidea* (Mas Cotek)

*Ficus deltoidea* (FD) is an evergreen plant native to the Malayan Archipelago [184]. A wide range of chemical compounds, including terpenoids, aliphatic groups, moretenol, lupeol, luteolin, rutin, quercetin, naringenin, vitexin, and isovitexin, have been isolated and characterized from FD [185,186,187,188]. FD has been shown to have antinociceptive, antimelanogenic, antioxidant, antiphotoaging, antibacterial, and antiulcerogenic properties, and is used in diabetes and inflammation conditions [189]. A study on BPA-induced female reproductive toxicity conducted on experimental animals, showed improvement in the BPA-induced uterine abnormalities, and enhanced the expression of ERα and ERβ and the immunity gene C3 [190]. In another study, FD exerted a preventive role against BPA-induced toxicity on the ovaries in rats [191].

#### 3.1.6. Sweet Potato (*Ipomoea batatas* L. Lam.)

Sweet potato (*Ipomoea batatas* L. Lam.), which originated from Central America, is one of the world’s most consumed crops [192]; it contains various bioactive compounds and nutrients, the sweet potato leaves, which are rich sources of antioxidants and can reduce malnutrition [192]. In addition to the antioxidant properties of sweet potato leaves, they are known for their minerals, vitamins, essential fatty acids, and dietary fibers [193,194]. A study has demonstrated that BPA administration was found to alter the structure and function of reproduction organs, eventually leading to infertility in rats. Eventually, an aqueous extract of *Ipomoea*
*batatas* was found to prevent the structural alterations and biochemical changes of the testis [193].

#### 3.1.7. *Quercus dilatata* Lindl. ex Royle

*Quercus dilatata* Lindl. ex Royle (QD) is one of the Quercus species, which is also known as Holly Oak and belongs to the family Fagaceae [195]. QD is considered a substantial source of bioactive metabolites. It is a traditional medicine for treating various diseases due to its biological activities, including antibacterial, hepatoprotective, anti-inflammatory, antioxidant, anticoagulant, and antidepressant properties [196,197,198]. Kazimi et al. have shown that the extracts of QD display protective effects against hepatotoxicity induced by BPA in rats [199]. This is the only report demonstrating QD’s ameliorative effect on BPA toxicity. They attributed this to the antioxidant and anti-inflammatory activities of QD.

#### 3.1.8. Tualang Honey

Tualang honey is found in Malaysia and is produced by wild honey bees feeding on the nectar of Tualang trees [200]. Compared with other Malaysian types of honey, Tualang honey had the highest content of flavonoids and phenolic compounds [201,202]. Its antioxidant and anti-inflammatory properties have been investigated in various diseases [203]. However, few studies have explored its ameliorative effect on BPA-induced toxicity. A study reported that the phytochemical properties of Tualang honey ameliorate the uterine disruption induced by BPA in rats [204]. In the same study, Tualang honey treatment restored the expression and distribution of ERα, ERβ, and complement 3 (C3) in the uterus [204]. In another study, authors found that treatment with Tualang honey in BPA-exposed rats improved the normal estrous cycle and protected the uterus by reducing its morphological abnormalities [205].

#### 3.1.9. Sesame Lignans

Sesame lignans are phytochemicals derived from sesame oil and have many biological properties [206]. They can be categorized into two types, inherent lignans, i.e., (sesamolin, sesamin), and lignans mainly formed during the process of oil production, i.e., (sesaminol, sesamol, sesamolinol, pinoresinol, matairesinol, lariciresinol, and episesamin [207]. These antioxidant lignans exhibit several biological activities, including anti-cancerous, antihypertensive, hypocholesterolemic, and antibacterial [208]. Despite the many studies that have been conducted on its biological activities, only a few studies have been performed to investigate their ameliorative effects against BPA-induced conditions. Eweda et al. have shown that hepatotoxicity and cardiotoxicity induced by BPA administration in Wistar rats were significantly attenuated by co-administration of sesame lignans. These lignans were able to restore the integrity of the heart and liver via the improvement of endogenous antioxidants [209]. In addition, oral gavage of sesame oil, a source of sesame lignans, ameliorates hepatotoxicity and DNA damage induced by BPA in mice [187]. A similar study revealed the protective activity of sesame oil against BPA exposure-induced cardiac effects in rats [210].

#### 3.1.10. Propolis

Propolis is a natural resin-like material produced by honeybees, by mixing beeswax with their saliva and substances collected from exudates and buds [211]. Bees use propolis to construct and maintain their hives [212]. The beneficial effects of propolis on human health have been reported since ancient history. It has various biological and pharmacological activities such as anti-inflammatory, antioxidant, antibacterial, anticancer, and antifungal [213]. Due to these properties and several health benefits, propolis and its extracts have been applied in treating different diseases [214]. The protective effects of propolis supplementation against BPA toxicity have been investigated. A study showed that co-administration of propolis significantly ameliorated liver toxicity induced by BPA in freshwater fish [215]. In addition, feed intake and growth that were retarded by BPA have been significantly improved by propolis supplementation. Another study also found that co-supplementation of propolis minimizes BPA-induced structural changes in rat lungs [216]. These reports indicated that propolis can be used as a protective agent against BPA toxicity.

#### 3.1.11. *Nigella sativa* Oil

*Nigella sativa* oil (NO) is pressed from the seeds of *Nigella sativa*, a widely used medicinal plant with a wide spectrum of health benefits [217,218]. Since antiquity, *Nigella sativa* seeds and their oil have been used to treat various diseases [218]. The therapeutic benefits of *Nigella sativa* seeds are attributed to the essential oil that contains thymoquinone, thymohydroquinone, carvacrol, nigellidine, and thymol [217,219]. Hyperlipidemia and obesity induced by BPA in mice were significantly alleviated by the ingestion of NO [220]. NO has also improved the reproductive and hematological functions in BPA-treated female mice [221]. In addition, in rats, co-supplementation of NO and thymoquinone have alleviated the metabolic disorders induced by BPA [222]. Thymoquinone, the major bioactive ingredient in NO has improved liver functions after BPA administration in rats [223]. Thymoquinone supplementation ameliorated BPA-induced oxidative stress and improved the antioxidant enzymes in the liver functions [223]. These reports indicate that *Nigella sativa* (and its oil) is an effective and natural compound for the treatment of various diseases.

#### 3.1.12. Green Tea

Green tea is one of the most popular drinks consumed worldwide [224]. It is processed from the plant *Camellia sinensis,* the same plant from which black tea and Oolong tea are processed [224]. However, green tea is a non-fermented tea due to the different processing during manufacturing, which started by steaming fresh *Camellia sinensis* leaves to prevent fermentation, maintain a green color, and preserve compounds with healthy properties [225]. Many researchers have studied green tea (and its extracts) due to its potential for preventing and treating various diseases such as cancer, diabetes, obesity, and cardiovascular diseases [226]. These health benefits were attributed to the high content of antioxidants, and anti-inflammatory components that influence different metabolic pathways [226,227,228]. Green tea has many health benefits due to its high content of polyphenolic compounds, including phenolic acid, flavonoids, flavonols, and flavandiols [229]. Its preventive effect against BPA toxicity has also been reported. Mohsenzadeh et al. have shown that vascular toxicity induced by BPA was prevented by the co-supplementation of green tea extract, epigallocatechin gallate (EGCG), or vitamin E in rats [230]. Another experimental study in rats has shown that green tea extract or EGCG co-supplementation prevented metabolic disorders induced by BPA through improving insulin signaling pathways (Figure 2) and lipid metabolism, which are attributed to their anti-inflammatory, and antioxidant properties [231]. An in vitro and in silico study by Suthar et al. revealed that green tea extract significantly mitigated the oxidative stress induced by BPA on erythrocytes [232].

#### 3.1.13. Soybean

Soybean, also known as soya bean, is a legume of the pea family Fabaceae. Consumption of soybeans has been associated with many beneficial effects on diseases, including diabetes, cancer, cardiovascular diseases, and obesity [233]. In addition to the high nutritional content, soybeans contain isoflavones that exhibit antioxidant, anti-inflammatory, and antimicrobial properties. In NMRI mice, co-administration of soybean extract with BPA significantly reduced fasting blood glucose and malondialdehyde levels in mice. Soybean also showed protective activity against some of the negative effects of BPA by increasing total antioxidative capacity, HOMA- and serum insulin levels [234]. In addition, in rats, the concurrent consumption of a soy-based diet counteracts the anxiogenic behavior induced by BPA [235]. The same study reported that the beneficial effects of a soy diet against BPA exposure-associated anxiogenic effects were mediated through the downregulation of estrogen receptor beta (Esr2) and melanocortin receptor (Mc3r, Mc4r) expressions in the amygdala (Figure 2).

**Figure 2 molecules-27-05384-f002:**
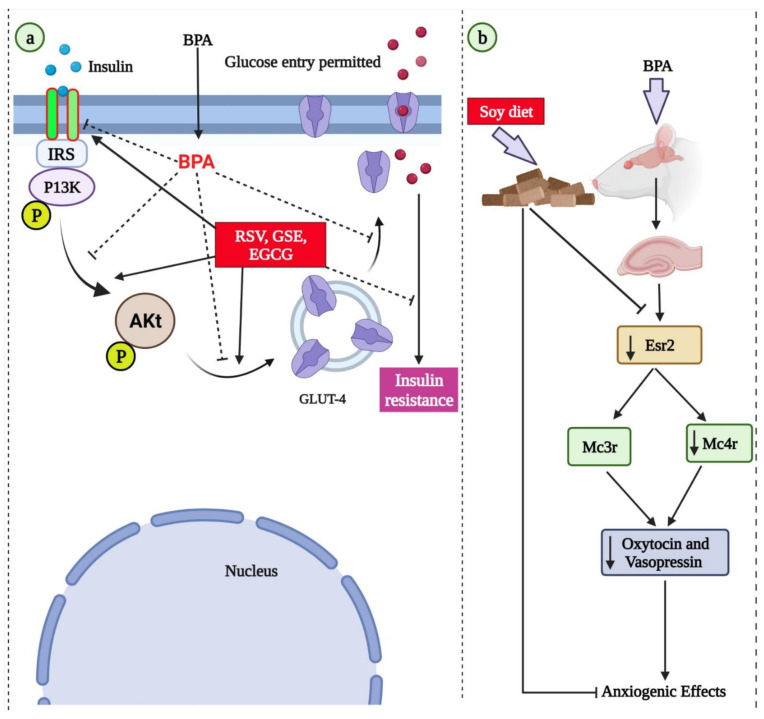
(**a**) The schematic representation of the insulin signaling promoting potential of resveratrol, grape seed extract [183], and epigallocatechin gallate (EGCG) [231]. (**b**) Developmental BPA exposure is associated with anxiogenic effects in juvenile rats via downregulating the expression of estrogen receptor beta (Esr2) and melanocortin receptors (Mc3r, Mc4r) in amygdala. The soy diet supplementation mitigated anxiogenic effects by upregulating these genes [235]. IRS: insulin receptor substrate; PI3K: phosphatidylinositol 3-kinase; Akt: protein kinase B.

#### 3.1.14. Pumpkin Seed Oil

Pumpkin seed oil (PSO) is rich in antioxidants and other nutritionally important nutrients such as vital fatty acids, phytosterols, amino acids, selenium, carotenes, etc. [236]. Tyrosol, ferulic acid, vanillic acid, vanillin, and luteolin are among the phenolic chemicals found in PSO. Furthermore, it has been proven to possess significant quantities of tocopherol, which translates to its antioxidant activity in minimizing lipid peroxidation [237]. Fawzy and colleagues investigated the beneficial effects of PSO on BPA exposure-induced toxicity in hepatic and testicular tissues of mice. PSO supplementation reduced DNA damage and improved histopathological changes in the liver and testes tissues exposed to BPA [238]. Additionally, administering PSO to mice before BPA treatment was the most effective method for reducing BPA’s negative effects, followed by administering PSO after BPA treatment [238].

#### 3.1.15. *Ginkgo biloba*

*Ginkgo biloba* (Gb) (Ginkgoaceae; Maidenhair tree) has recently received a lot of interest for its neuroprotective properties [239,240]. The two principal components responsible for its distinctive pharmacological impact were flavonoids and terpenoids [241]. Gb has been shown to have estrogenic [242], antioxidant [243], and adiponectin pro-secretory effects [244] capabilities, ensuring its flexibility as a learning and memory supplement. In an animal model of rats, researchers investigated the neuroprotective effects of Gb on BPA exposure-induced neurotoxicity and discovered that Gb pretreatment enhanced cognitive function, which might be due to increased hippocampus levels of estrogen-dependent biogenic amines [245]. Simultaneously, Gb was able to effectively reduce BPA-induced oxidative stress by increasing SOD activity and adiponectin levels while lowering nitric oxide and malondialdehyde levels. Finally, Gb reduced the histopathological damage caused by BPA and blocked the activation of NF-B and caspase-3 [245].

#### 3.1.16. Ginseng

Ginseng belongs to the *Panax* genus of the Araliaceae family. In folk medicine, it is used to heal various health conditions. There are various Panax species, among them the *Panax ginseng* (Korean ginseng), *Panax notoginseng* (Chinese ginseng), and *Panax quinquefolius* (American ginseng) are widely used species. These species differ in their ginsenoside contents. Ginsenoside Rf is one of the important compounds unique to Korean ginseng. The Korean Red Ginseng (KRG) is the end product of processed Korean ginseng and is well recognized by the Korean Food and Drug Administration [246]. KRG is effective against cancer, atherosclerosis, and neurodegenerative diseases through anti-oxidant and anti-inflammatory properties [247,248,249,250]. In an in vitro study, KRG showed anti-inflammatory effects on BPA-treated A549 lung cells by reducing the production of reactive oxygen species and altering the NF-κB activation and COX-2 expression [251]. In another in vitro study, ginseng prevented BPA-induced apoptosis by upregulating the anti-apoptosis systems [252]. The effect of KRG supplementation on BPA-induced changes in the liver and uterus of ovariectomized (OVX) animal model has also been studied, which concluded that KRG is protective against BPA-induced inflammatory responses and chemotaxis in ovariectomized (OVX) mice [253]. In another study of the OVX mice animal model, KRG inhibited BPA-induced altered lipid metabolism by regulating the lipid metabolic process-related genes [254]. In rats, ginseng alleviated the BPA-induced reproductive toxicity during pregnancy by reversing abnormal testosterone and progesterone levels to normal [255].

#### 3.1.17. *Murraya koenigii*

*Murraya koenigii* (MK) (curry-leaf tree) belongs to the Rutaceae family. In Indian traditional medicine (Ayurveda), it is used to treat various diseases. The plant leaves were used as appetizers, analgesics, anthelmintics, and digestives. They effectively treat various conditions, including piles, itching, edema, inflammation, fresh cuts, dysentery, and bruises [256]. In experimental studies, its antimicrobial, nephroprotective, anti-oxidative, hepatoprotective, antifungal, and anti-inflammatory properties have been reported [257,258,259]. Its hydroethanolic leaf extract possesses carbazole alkaloids, which have antioxidant and radicle scavenging activities [260]. In BALB/c mice, MK extract treatment significantly alleviated the BPA-induced testicular damage as well as apoptosis. In this study, MK extract treatment improved the sperm parameters, germ cell number, and antioxidant enzyme activity, and also increased Bcl-2 and decreased caspase-9 and caspase-3 gene expression in BPA-exposed mice [261].

#### 3.1.18. *Asparagus officinalis*

*Asparagus officinalis* (AO) has been traditionally used in many parts of Europe and Asia as traditional medicine and is consumed in salads, vegetable dishes, and soups [262]. AO has been reported to exhibit antifungal, antimutagenic, anti-inflammatory, and diuretic properties [263]. One of the complications of diabetes mellitus, i.e., diabetic nephropathy, has been effectively treated with AO [264]. Active compounds, such as flavonoids and polyphenols, were reported to be responsible for the beneficial antioxidant effect of AO [265]. In a study on the protective effects of AO against BPA-induced liver and kidney tissue damage in rats, it was shown that AO co-supplementation significantly prevented BPA-induced toxicity in these tissues, potentially by preventing GSH depletion, suppressing lipid peroxidation, and promoting antioxidative capacities [100].

#### 3.1.19. *Aloe vera* (*Aloe barbadensis* Miller)

*Aloe vera* belongs to the Xanthorrhoeaceae family. It is a perennial green herb consisting of stiff gray-green lance-shaped leaves with a gel in a central mucilaginous pulp [266]. Traditionally, it is used to treat dermal problems. Additionally, it possesses anti-inflammatory, anticancer, antioxidant, antidiabetic, and antihyperlipidemic properties [267]. It has been shown to promote spermatogenesis [268]. *Aloe vera* gel extract co-supplementation in rats significantly ameliorated BPA-testicular toxicity attributed to its antioxidant properties [269].

#### 3.1.20. *Tribulus terrestris* L.

*Tribulus terrestris* L. (TT) is an annual creeper generally available in India, South Africa, Australia, and Europe. It belongs to the Zygophyllaceae family. Its beneficial effects against various diseases have been mentioned in the traditional medicine of China, Indian Ayurveda, and Bulgaria [270]. It is a well-known, commercially available natural herbal product in the form of powder, capsules, and tea. It is rich in vitamins, flavonoids, alkaloids, tannins, steroids, spooning, unsaturated fatty acids, etc. [271]. Both in vitro and in vivo studies have demonstrated its anti-inflammatory, anti-hyperglycemic, antioxidant, and antibacterial properties. In a recent clinical study, TT treatment positively affected male sexual dysfunction [272]. In rats, TT supplementation significantly prevented BPA exposure-induced histopathological changes in the testes and reduced testosterone hormone levels [273].

### 3.2. Natural Compounds

#### 3.2.1. Resveratrol

Resveratrol (RSV) is a biologically active polyphenol compound produced in plants that are exposed to ionizing or infectious radiation [274]. It was first isolated from white hellebore roots, and since then has been identified in almost 70 plant species. Red grapes possess high amounts of RSV, probably as a result of *Vitis vinifera* (grapevine) response to fungal infection. It is also present in red wine. Commercially, it is synthesized by yeasts called *Saccharomyces cerevisiae* [275,276,277]. RSV butyrate esters (RBE) are derived from RSV and butyric acid, and they present higher bioavailability but similar bioactivity to RSV. Perinatal BPA exposure increased body weight, lipid accumulation, and blood lipid levels, and also affected the intestinal microbiota in the female offspring rats [278]. Perinatal RBE supplementation ameliorated the BPA-induced obesity in female offspring rats. Further RBE reduced the Lactobacillus abundance, and Firmicutes/Bacteroidetes (F/B) ratio, and increased the S24-7 abundance. It also reduced fecal acetate levels [278]. In another study, RSV and mesenchymal stem cell supplementation, either alone or in combination in rats, significantly ameliorated the BPA-induced uterine endometrial damage in rodents. These effects were mediated through various molecular mechanisms of regulation, such as promoting gonadal hormone synthesis, anti-fibrotic changes, reducing oxidative stress markers, and apoptosis-related genes [279]. RSV treatment has been shown to be effective against BPA-induced cellular toxicity in rat salivary glands [280]. In a recent study, RSV was a potential candidate for preventing perinatal BPA-induced atherosclerosis lesion formation in the adult offspring Apo E mice [155]. In rats, RSV treatment was effective in preventing liver histopathological changes caused by BPA exposure [281].

In ovarian cancer cells, RSV prevented cell proliferation by suppressing the cross-talk between estrogen receptor α and insulin growth factor-1 receptor signaling pathways [282]. In another in vitro study, RSV treatment reduced calcium levels and TRPM2 channel currents in BPA-exposed cortical collecting duct cells in the kidney [283]. It also ameliorated the BPA-promoted membrane depolarization of mitochondria and apoptosis [283]. Furthermore, in rats, RSV was also shown to be a potential candidate for preventing BPA and high-fat-diet-induced developmental programming of hypertension in adult offspring [160]. These beneficial effects against BPA and high-fat-diet-induced developmental programming of hypertension were found to be mediated through AhR signaling pathways (Figure 1). In a recent study involving rats, RSV butyrate ester treatment attenuated the perinatal BPA-induced liver damage by suppressing oxidative stress and modulating the gut-microbiota, specifically Adlercreutzia and S24-7 [284]. In mice, RSV treatment was also shown to inhibit BPA-induced male reproductive toxicity [285]. Another study involving rats showed a protective effect of RSV against BPA exposure-induced damage in oral mucosa and the tongue [286].

#### 3.2.2. Luteolin

Luteolin (3′,4′,5,7-tetrahydroxyflavone) is a naturally occurring flavonoid. Luteolin is normally found in glycosylated forms in green pepper, basic perilla leaf, seed, celery, honeysuckle bloom, and chamomile blossom [287,288,289]. Research suggests that luteolin has potent anti-inflammatory, antidiabetic, antitumor, antiapoptotic, hepatoprotective, and chemoprotective properties [290,291,292]. Having a C (6-3-6) structure, luteolin belongs to the flavone group of flavonoids. The hydroxyl moieties and double bonds are basic structures incorporated into luteolin that are known to be linked to its biochemical and biological activities [293].

In a study by Adesanoye and colleagues, BPA exposure induced antioxidant-oxidative stress imbalance and behavioral deficits in Drosophila melanogaster flies [294]. Luteolin treatment increased the survival rate and enhanced antioxidant markers in flies. It was also revealed that luteolin could ameliorate the BPA exposure-induced fatty degeneration, behavioral changes, oxidative stress, cell viability reduction, and eclosion rate of flies [294]. Oral administration of luteolin reduced BPA exposure-induced kidney abnormalities, blood urea nitrogen, creatinine, and serum uric acid levels in rats [293]. In addition, luteolin reduced inflammatory mediators caused by BPA, such as interleukin 1 beta, tumor necrosis factor-alpha, and interleukin 6. Luteolin treatment also enhanced heme oxygenase 1 (HO-1) as well as nuclear factor-like 2 (Nrf2) expressions, indicating its nephroprotective role through the Nrf2/antioxidant response element (ARE)/HO-1 pathway [293] (Figure 3).

#### 3.2.3. Lycopene

Lycopene is a natural lipophilic unsaturated carotenoid present in red-colored fruits and vegetables, particularly tomatoes. It exerts strong antioxidant and free radical scavenging properties [295]. Its anti-oxidative capacity in ameliorating the toxic effects of different compounds has been tested previously [296,297,298]. In experimental animal studies, lycopene treatment was effective in preventing BPA exposure-induced lung injury [299], hepatic toxicity [300], and reproductive toxicity [301] through its anti-apoptotic, antioxidant, and anti-inflammatory effects. In Wistar rats, lycopene treatment reversed the BPA exposure-induced metabolic changes, such as increased total antioxidant capacity, altered lipid profile, increased glucose intolerance and insulin resistance, and decreased thyroid hormone levels [302]. Lycopene also ameliorates BPA exposure-induced memory impairment and hippocampal neuronal damage in rats [303]. The molecular mechanism of lycopene against BPA exposure-induced neurotoxicity is shown in Figure 3.

**Figure 3 molecules-27-05384-f003:**
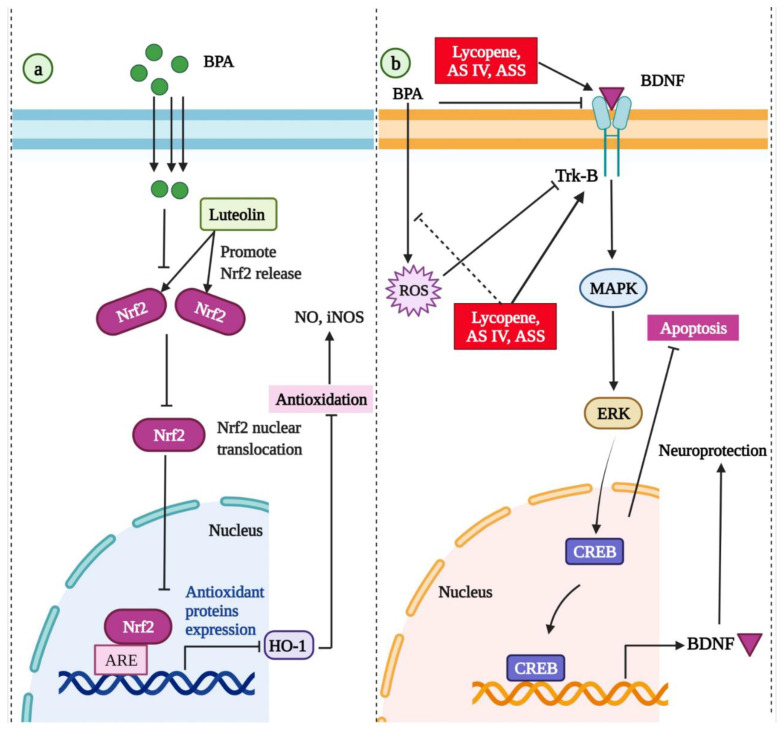
(**a**) The molecular mechanism of antioxidant effects of luteolin against BPA-induced renal toxicity through Nrf2/ARE/HO-1 pathway modulation [293]. (**b**) The molecular mechanism of lycopene [303], *Astragalus spinosus* saponins (ASS), and Astragaloside IV (AS IV) [304] against BPA exposure-induced neurotoxicity. Nrf2: nuclear factor-like 2; ARE: antioxidant response element; HO-1: heme oxygenase 1; BDNF: brain-derived neurotrophic factor; TrKb: tyrosine receptor kinase B; ERK: extracellular signal-regulated kinases; MAPK: mitogen-activated protein kinase; CREB: cAMP response element-binding protein.

#### 3.2.4. *Astragalus spinosus saponins* and *Astragaloside IV*

Astragaloside IV (AS IV) is the core bioactive component of *Astragalus spinosus*. It is recognized by the presence of 3-, 6-, and/or 25-coupled glucose moieties [305]. AS IV is known to have various biological activities, such as anti-inflammatory, antiapoptotic, and antilipolytic properties [306]. AS IV supplementation lowered the nitric oxide (NO) production in brain tissues and improved the blood–brain barrier’s (BBB) permeability following ischemia [307]. Saponins are fundamental biologically active components of *Astragalus spinosus* [308]. *A. spinosus* has potential neuroprotective effects linked to its antioxidant, anti-depressive, and anti-apoptotic roles [309]. In rats, administration of ASIV or *A. spinosus* saponins prevented BPA exposure-induced neurotoxicity by reducing oxidative stress and restoring the expression of brain-derived neurotrophic factor and N-methyl-D-aspartate receptors [304]. The role of *A. spinosus* saponins and ASIV on brain-derived neurotrophic factor expression regulation is shown in Figure 3. Both treatments also improved histological changes caused by BPA exposure in various regions of the brain [304]. In another study in rats, administration of AS IV and *A. spinosus* saponins reversed the BPA-induced anxiogenic and depressive-like behaviors [310]. Furthermore, these compounds improved memory, restored serotonin, dopamine, and monoamine oxidase levels, and normalized the expression of Tph2 mRNA [310].

#### 3.2.5. Naringin

Naringin (4′,5,7-Trihydroxyflavanone-7-Rhamnoglucoside) is a well-known flavanone glycoside found in various plant species, predominantly citrus fruits. Other sources of naringin include cacao, cherries, and tomatoes [311]. Naringin is known for its pharmacological benefits against inflammation, tumors, oxidative stress, and apoptosis. It is effective in preventing cardiovascular diseases, neurodegenerative disorders, genetic damage, and bone loss [312,313]. In a recent study, BPA exposure-induced cardiac toxicity by increasing the levels of aspartate aminotransferase, lactate dehydrogenase, creatine kinase–MB, triglyceride, lipid peroxidation, as well as decreasing the levels of glutathione, superoxide dismutase, catalase, and glutathione peroxidase. Interestingly, in Wistar rats treatment with higher doses of Naringin significantly ameliorated the BPA-induced cardiac toxicity parameters [314]. Mahdavinia et al. demonstrated that co-administration of Naringin ameliorated the BPA-induced cognitive dysfunction and oxidative damage in Wistar rats [315].

#### 3.2.6. Taurine

Taurine (2-aminoethanesulfonic acid) is a natural amino acid found in various mammalian tissues, including their reproductive systems [316]. Taurine has a primary role as an antioxidant in the body [317]. It exhibits antioxidant, anticancer, antitumor, and antifibrotic activities [318,319]. Important biological functions of taurine include membrane stability, sperm motility, glucose balance, and the development of the CNS [320]. A study has shown that BPA exposure in NMRI mice impaired spermatozoa viability, and motility. Pretreatment with taurine suppressed mitochondrial oxidative stress and improved sperm motility and viability [321]. In another study, the neuroprotective effects of taurine against BPA-induced neurotoxicity were investigated. Taurine co-administration ameliorated the BPA exposure-induced behavioral changes of zebrafish by reducing oxidative stress [322].

#### 3.2.7. Quercetin

Quercetin (3, 3′, 4′, 5, 7-pentahydroxyflavone) is a flavonoid found in plants with health benefits [323]. Quercetin is usually found in fruits, vegetables, and cereal [324]. Quercetin is widely known for its properties against allergy, inflammation, ischemia, cancer, and viral infections [325]. Mahdavinia et al. highlighted the benefits of quercetin in a BPA-induced hepatotoxicity rat model by preventing mitochondrial damage and reducing oxidative stress [108]. Likewise, the findings of Shirani et al. showed that quercetin could reduce the toxic effects of BPA in isolated mitochondria from rat kidney tissues [326]. Quercetin treatment in rats also ameliorated the toxic effects of BPA on the testis and epididymis, the impairment of spermatogenesis, and the imbalance in hormonal levels and lipid profile [327]. Quercetin has also been shown to improve neurobehavioral response in zebrafish [328], reduce oxidative stress in human erythrocytes [329], and protect against liver and kidney damage in mice [330].

#### 3.2.8. Genistein

Genistein is a natural isoflavone and phytoestrogen of the soybean with a wide range of pharmacological properties [331]. Genistein has several biological effects, such as antioxidant, antibacterial, and anti-inflammatory activities. Thus, it has been used to treat various diseases, including osteoporosis, diabetes, lipid metabolism, and cardiovascular disease [332]. Moreover, the protective effects of genistein have been examined against BPA-induced toxicity in different organs. Maternal genistein supplementation in rats attenuated the adverse effects of gestational BPA exposure on early and late prostate development through the alteration of epithelial cell proliferation and the expression of androgen receptors [333]. Yakimchuk et al. suggested that phytoestrogen supplementation for lymphoma patients prevented lymphoid malignancies [334]. Surprisingly, prepubertal exposures to BPA and genistein modified the expression of a large number of proteins in rat sera at postnatal days 21 and 35 [335].

#### 3.2.9. Curcumin

Turmeric (*Curcuma longa*), often known as ‘curcuma domestica’ belongs to the Zingiberaceae family. It is a herbaceous perennial plant [336]. Despite the fact that it contains over 300 active components, the principal biologically active component establishing the basis for the therapeutic capabilities of this plant is a substance taken from its root called curcumin [337], which is a polyphenolic molecule [338]. It has a wide range of pharmacological effects, including antioxidant, antiproliferative, anticancer, immunomodulatory, antimicrobial, and anti-inflammatory properties [339]. Curcumin has also been demonstrated to possess renoprotective [340], neuroprotective [339], and cardioprotective [340] properties. Uzunhisarcikli and Aslanturk investigated the potential therapeutic effects of curcumin and taurine by targeting the BPA-induced liver injury in rats [341]. Both curcumin and taurine reduced the negative effects of BPA in the liver by reducing lipid peroxidation products and oxidative stress [341]. A previous study that investigated the protective effects of turmeric against BPA exposure-induced genotoxicity in Wistar NIN rats revealed that turmeric treatment decreased serum malondialdehyde and urinary 8-hydroxy-2′-deoxyguanosine levels. In addition, turmeric co-administration decreased the formation of micronuclei and DNA migration in hepatic and renal tissues [342]. Curcumin has been shown to reduce BPA-induced oxidative stress and histopathological alterations in Wistar rats’ testis and cardiac tissues [343,344]. The summary of the beneficial effects of various natural compounds against BPA-induced adverse effects of experimental animal models is shown in Table 1.

A schematic diagram showing the various NPs evaluated for their potential ameliorating roles against specific BPA-induced toxicity is presented in Figure 4.

## 4. Conclusions

Exposure to BPA in everyday life is practically unavoidable. BPA is an estrogenic endocrine disrupting chemical, and the adverse effects on different body organs are a cause of concern. Therefore, there is a need to curb exposure to BPA in both adults and pregnant women. Even though potential molecular mechanisms underlying BPA-induced toxicity have been investigated, there is currently no specific targeted treatment for BPA-induced toxicity in humans. Hence, there is a need to develop a therapeutic drug that antagonizes BPA-induced toxicity. In the past, NPs have significantly contributed to drug discovery to treat diseases, particularly cancer and infectious diseases. An example is a fungus-derived fingolimod drug, which the FDA approved for the treatment of multiple sclerosis [345,346]. One of the major advantages of NPs in drug discovery is that they confer multiple “targets” by targeting more than one signaling pathway [347]. In addition, most NPs showed beneficial effects through their antioxidant, anti-inflammatory, and anti-apoptotic properties. The plant extracts, including *P. integerrima*, green tea, soy-rich diet, Gb, KRG, and ginseng, seem to be the most promising in alleviating BPA-induced toxicity. However, the active compounds in these extracts need to be explored. On the other hand, natural compounds, such as RSV, luteolin, lycopene, AS IV, genistein, and curcumin are found to be most promising in mitigating BPA toxicity. In the future, more research should be conducted to explore the complex network of molecular mechanisms to precisely understand the roles of NPs. The main challenges of NP-based drug development are attributed to its poor bioavailability and determining the optimal dose. Hence, further pharmacokinetic studies in clinical settings are warranted.

## Figures and Tables

**Figure 1 molecules-27-05384-f001:**
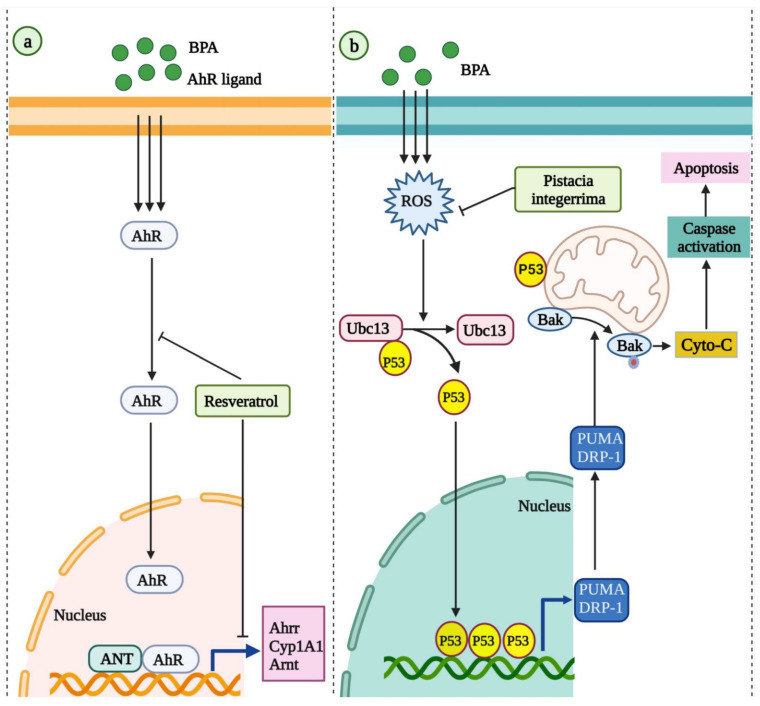
(**a**) The antioxidant effects of resveratrol against BPA and high-fat-diet-induced developmental programming of hypertension through the AhR signaling pathways [160]. (**b**) The molecular mechanisms involved in the anti-apoptotic effects of *Pistacia integerrima* against BPA exposure-induced toxicity in heart tissue [157]. AhR: aryl hydrocarbon receptor; Cyp1a1: cytochrome P450 Cyp1a1; Cyto C: cytochrome C; PUMA: P53 upregulated modulator of apoptosis; Drp1: dynamin-related protein 1; Ubc13: ubiquitin-conjugating enzyme variant.

**Figure 4 molecules-27-05384-f004:**
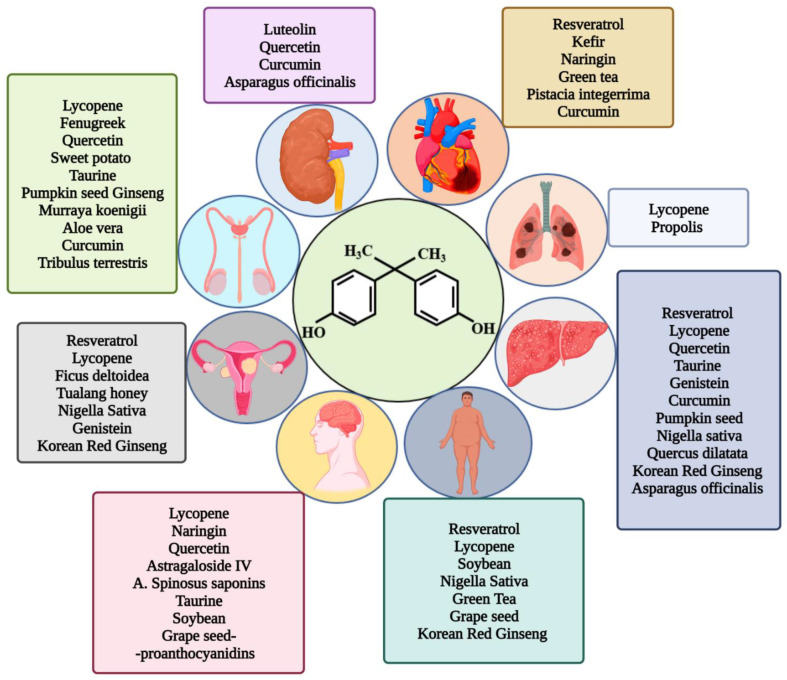
Schematic diagram showing the various NPs evaluated for their potential ameliorating roles against specific BPA-induced toxicity.

**Table 1 molecules-27-05384-t001:** Summary of experimental studies evaluating the ameliorative potential of natural products and natural compounds against BPA exposure-induced toxicity.

Author; Year	Animal Model	BPA Dose	Natural Product/Natural Compound and Its Dose	BPA Induced Toxicity	Mechanism of Actions
Ishtiaq et al., 2020 [157]	Sprague Dawley rats	100 µg/kg B.wt/day	*Pistacia integerrima*— 200 mg/kg B.wt/day	Cardiotoxicity	Neutralizing the oxidative stress through Ubc13/p53 pathway
Kaur, S., and Sadwal, S. 2020 [171]	Mice (BALB/c)	1 mg/kg B.wt/day	Fenugreek seed extract—200 mg/ kg B.wt/day	Testicular damage	-Antioxidant effects
Friques et al., 2020 [174]	Wistar rats	100 μg/kg B.wt/day	Kefir—0.3 mL/100 g B.wt/day	Hypertension and vascular toxicity	-Antioxidant effects -Increasing NO bioavailability
Abdou et al., 2022 [179]	Wistar rats	50 mg B.wt/day/kg	Grape seed proanthocyanidins—200 mg/kg B.wt/day	Neurotoxicity	-Anti-inflammatory effects -Antioxidant effects
Zaid et al., 2021 [190]	Sprague Dawley rats	10 mg/kg B.wt/day	*Ficus deltoidea*—100 mg/kg B.wt/day	Female reproductive toxicity (Uterus)	NA
Zaid et al., 2018 [191]	Sprague Dawley rats	10 mg/kg B.wt/day	*Ficus deltoidea*—100 mg/kg B.wt/day	Female reproductive system (ovary)	NA
Revathy et al., 2017 [193]	Sprague Dawley rats	200 mg/kg B.wt/day	*Ipomoea batatas*—400 mg/kg B.wt/day	Male reproductive toxicity	NA
Kazmi et al., 2018 [199]	Sprague Dawley rats	25 mg/kg B.wt/day	*Quercus dilatata* Lindl. ex Royle—300 mg/kg B.wt/day	Hepatotoxicity	Antioxidant effects
Mohamad Zaid et al., 2015 [204]	Sprague Dawley rats	10 mg/kg B.wt/day	Tualang honey—200 mg/kg B.wt/day	Uterine toxicity	-Normalizing ERα, ERβ, and C3 expression and distribution -Reducing lipid peroxidation
Zaid et al., 2014 [205]	Sprague Dawley rats	10 mg/kg B.wt/day	Tualang honey—200 mg/kg B.wt/day	Ovarian toxicity	Antioxidant effects
Eweda et al., 2020 [209]	Albino Wistar rats	30 mg/kg B.wt/day	Sesame lignans—20 mg/kg B.wt/day	Hepatotoxicity and cardiotoxicity	-Antioxidant effects -Improving lipid profile
Abo et al., 2020 [210]	Sprague Dawley rats	25 and 50 mg/kg B.wt/day	Sesame oil—10 mL/kg B.wt/day	Cardiotoxicity	Antioxidant effects
Soliman et al., 2021 [216]	Albino rats	500 mg/kg B.wt/day	Propolis—50 mg/kg B.wt/day	Lung injury	Anti-inflammatory and antioxidant effects
Sujan et al., 2019 [220]	Swiss albino mice	50 mg/kg B.wt/day	Nigella Sativa oil— 1 mL/kg B.wt/day	Hyperlipidemia and obesity	Antioxidant effects
Sujan et al., 2020 [221]	Swiss albino mice	50 mg/kg B.wt/day	Nigella Sativa oil— 1 mL/kg B.wt/day	Blood and reproductive organ	Antioxidant effects
Fadishei. et al., 2021 [222]	Albino Wistar rats	10 mg/kg B.wt/day	Nigella Sativa oil— 21, 42, 84 μL/kg B.wt/day Thymoquinone—0.5, 1, 2 mg/kg B.wt/day	Metabolic disorder	Antioxidant effects
Abdel-Wahab et al., 2014 [223]	Sprague Dawley (SD) rats	10 mg/kg B.wt/day	Thymoquinone—10 mg/kg B.wt/day	Hepatoxicity	Antioxidant effects
Mohsenzadeh et al., 2021a [230]	Wistar rats	10 mg/kg B.wt/day	Green tea— 25, 50, and 100 mg/kg B.wt/day Epigallocatechin gallate—10, 20, and 40 mg/kg/day	Vascular toxicity	Antioxidant effects
Mohsenzadeh et al., 2021b [231]	Albino Wistar rats	10 mg/kg B.wt/day	Green tea— 25, 50, and 100 mg/kg B.wt/day Epigallocatechin gallate—10, 20, and 40 mg/kg/day	Metabolic disorders	-Anti-inflammatory effects -Regulating the metabolism of lipids -Improving insulin signaling pathways
Veissi et al., 2018 [234]	NMRI mice	100 μg/kg B.wt/day	Soy extract— 60, 150 mg/kg B.wt/day	Metabolic disorder	Antioxidant effects
Patisaul et al., 2012 [235]	Wistar rats	1 mg/L	Soy rich diet	Anxiogenic behavior	Estrogen receptor beta, melanocortin receptors, oxytocin/vasopressin signaling pathways
Fawzy et al., 2018 [238]	Swiss albino mice	50 mg/kg B.wt/day	Pumpkin seed oil—1 mL/kg B.wt/day	DNA damage in the liver and testes	Decreasing DNA damage
El Tabaa et al., 2017 [245]	Wistar rats	250 mg/kg B.wt/day	Ginkgo biloba extract— mg/kg B.wt/day	Neurotoxicity	-Increasing biogenic amines release -Antioxidant effects adiponectin pro-secretory effects
Lee et al., 2020 [253]	CD-1 mice	200 mg/kg B.wt/day	Korean red ginseng—1.2 g/kg/day	Inflammation in liver and uterus	Anti-inflammatory effects
Park et al., 2020 [254]	ICR mice	800 mg/kg B.wt/day	Korean red ginseng—1.2 g/kg/day	Increased lipid profile	Regulating lipid metabolic process-related genes
Saadeldin et al., 2018 [255]	Albino rats	150 mg/kg B.wt/day	Ginseng—200 mg/kg B.wt/day	Reproductive toxicity	Modulating mRNA transcripts of STAR, HSD17B3, and CYP17B, via AKT/PTEN pathway
Kaur et al., 2020 [261]	BALB/c mice	1 mg/kg B.wt/day	*Murraya koenigii*—200 mg/kg B.wt/day	Testicular toxicity	-Antioxidant effects -Antiapoptotic effects
Poormoosavi et al., 2018 [100]	Wistar rats	10 mg/kg B.wt/day	*Asparagus officinalis*— 200 mg/kg B.wt/day	Hepatic and renal toxicity	Antioxidant effects
Behmanesh et al., 2018 [269]	Wistar rats	20 μg/kg B.wt/day	*Aloe vera* gel—300 mg/kg B.wt/day	Testicular toxicity	Antioxidant effects
Munir et al., 2017 [273]	Sprague Dawley rats	25 mg/kg B.wt/day	*Tribulus terrestris* L. —20 mg/kg B.wt/day	Testicular toxicity	NA
Sirasanagandla et al., 2022 [155]	Apo E mice	1 μg/ml	Resveratrol—20 mg/kg B.wt/day	Atherosclerosis	Autophagy modulation
Rameshrad et al., 2018 [182]	Albino Wistar rats	35 mg/kg B.wt/day	Resveratrol—100 mg/kg B.wt/day Grape Seed Extract—3, and 12 mg/kg B.wt/day	Vascular toxicity	Antioxidant effects
Rameshrad et al., 2019 [183]	Wistar rats	35 mg/kg B.wt/day	Resveratrol—25, 50, and 100 mg/kg B.wt/day Grape seed extract—3, 6, 12 mg/kg B.wt/day	Metabolic syndrome and insulin resistance	-Promoting insulin signaling -Increasing *ABCG*8 expression in the liver -Antioxidant activity
Shih et al., 2021 [278]	Sprague Dawley rats	50 μg/kg B.wt/day	Resveratrol butyrate esters— 30 mg/kg B.wt/day	Obesity	Modulatory activity in intestinal microbiota
Fouad et al., 2021 [279]	Wistar rats	20 mg/kg B.wt/day	Resveratrol— 20 mg/kg B.wt/day	Uterine damage	-Antioxidant activity -Antiapoptotic effects
Cetin et al., 2021 [280]	Wistar albino rats	130 mg/kg B.wt/day	Resveratrol—100 and 200 mg/kg/day Apigenin—100 and 200 mg/kg B.wt/day	Salivary gland cytotoxicity	-Antioxidant effects -Antiapoptotic effects
Bordbar et al., 2021 [281]	Sprague Dawley rats	50 mg/kg B.wt/day	Resveratrol—100 mg/kg B.wt/day	Hepatotoxicity	NA
Hsu et al., 2019 [160]	Sprague Dawley rats	50 μg/kg B.wt/day	Resveratrol—50 mg/L	Developmental programming of hypertension	-Increasing NO bioavailability -Antioxidant effects -Suppressing the AHR signaling pathway
Liao et al., 2021 [284]	Sprague Dawley rats	50 μg/kg B.wt/day	Resveratrol butyrate esters—30 mg/kg B.wt/day	Hepatic toxicity	-Antioxidant effects -Modulating gut microbiota
Rahmani-Moghadam et al., 2022 [286]	Sprague Dawley rats	50 mg/kg B.wt/day	Resveratrol—100 mg/kg B.wt/day	Oral mucosa and tongue toxicity	NA
Alekhya Sita al., 2019 [293]	Wistar rats	250 mg/kg B.wt/day	Luteolin—100 and 200 mg/kg B.wt/day	Nephron toxicity	Nrf2/ antioxidant response element (ARE)/HO-1 pathway regulation
Adesanoye et al., 2020 [294]	Drosophila melanogaster (Canton-S strain)	0.05 mM	Luteolin—150 and 300 mg/kg B.wt/day	Oxidative stress, locomotor deficit, reduction in offspring emergence rate, cell viability, inhibition of acetylcholinesterase activity	-Antioxidant and chemo-preventive properties
Faheem et al., 2021 [299]	Albino Wistar rats	50 mg/kg B.wt/day	Lycopene—10 mg/kg B.wt/day	Lung injury	-Anti-inflammatory effects -Antioxidant effects -Antiapoptotic effects
Abdel-Rahman et al., 2018 [300]	Wistar rats	10 mg/kg B.wt/day	Lycopene—10 mg/kg B.wt/day	Hepatotoxicity	-Antioxidant effects -Antiapoptotic effects
Ma et al., 2018 [301]	Kunming mice	500 mg/kg B.wt/day	Lycopene—20 mg B.wt/day/kg	Reproductive toxicity	NA
Elgawish et al., 2020 [302]	Wistar rats	10 mg/kg B.wt/day	Lycopene—10 mg/kg B.wt/day	Metabolic syndrome	-Antioxidant effects -Anti-inflammatory effects
El Morsy et al., 2020 [303]	Albino rats	50 mg/kg B.wt/day	Lycopene—10 mg/kg B.wt/day	Hippocampal neurotoxicity and defective memory function	-Antioxidant effects -Activation of MAPK/ERK pathway -Antiapoptotic effects
Essawy et al., 2021 [304]	Sprague Dawley rats	125 mg/kg B.wt/day	Astragaloside IV—80 mg/kg B.wt/day *A. spinosus* saponins-100 mg/kg B.wt/day	DNA damage and Neurotoxicity	-Antioxidant effects -Anti-inflammatory and anti-apoptotic effects -Reducing DNA damage -Regulating the BDNF and NR2A and NR2B gene expression
Abd Elkader et al., 2021 [310]	Sprague Dawley rats	125 mg/kg B.wt/day	Astragaloside IV—80 mg/kg B.wt/day *A. spinosus* saponins-100 mg/kg B.wt/day	Long-lasting anxiety-like behavior and depression in schizophrenia	-Neuroprotective activity
Khodayar et al., 2020 [314]	Wistar rats	50 mg/kg B.wt/day	Naringin—40, 80, and 160 mg/kg B.wt/day	Cardiotoxicity	-Lipid-lowering properties -Antioxidant effects -Decreasing lipid peroxidation
Mahdavinia et al., 2019 [315]	Wistar rats	50 mg/kg B.wt/day	Naringin—40, 80, and 160 mg/kg B.wt/day	Cognitive impairment and oxidative damage	-Antioxidant and neuroprotective effects
Rezaee-Tazangi et al., 2020 [321]	NMRI mice	0.8 mmol/mL	Taurine—5, 10, 30, and 50 µmol/L	Mitochondrial toxicity and impaired sperm quality	Antioxidant effects
Mahdavinia et al., 2019 [108]	Wistar rats	250 mg/kg B.wt/day	Quercetin—75 mg/kg B.wt/day	Hepatotoxicity (liver)	-Antioxidant effects -Preventing mitochondrial damage
Pradhan et al., 2021 [322]	Zebrafish	17.52 μM	Taurine—63.9233 μM	Neurotoxicity	Antioxidant effects
Shirani et al., 2019 [326]	Wistar rats	250 mg/kg B.wt/day	Quercetin—75 mg/kg B.wt/day	Nephrotoxicity (through uric acid and creatinine)	Antioxidant effects
Jahan et al., 2016 [327]	Sprague Dawley rats	50 mg/kg B.wt/day	Quercetin—50 mg/kg B.wt/day	Testicular toxicity	NA
Sahoo et al., 2020 [328]	Zebrafish	17.52 μM	Quercetin—2.96 μM	Neurotoxicity	Antioxidant effects
Sangai et al., 2014 [330]	Swiss albino mice	120 and 240 mg/kg B.wt/day	Quercetin—60 mg/kg B.wt/day	Hepatotoxicity and nephrotoxicity	Antioxidant effects
Bernardo et al., 2015 [333]	Sprague Dawley rats	25 and 250 μg/kg B.wt/day	Genistein— 5.5 mg/kg B.wt/day	Reproductive organs	Antitumor effects
Betancourt et al., 2014 [335]	Sprague Dawley rats	250 μg/kg B.wt/day	Genistein— 250 mg/kg B.wt/day	Cancer	Anticancer and chemoprotective effects
Uzunhisarcikli and Aslanturk, 2019 [341]	Wistar rats	130 mg/kg B.wt/day	Curcumin—100 mg/kg/day Taurine—100 mg/kg B.wt/day	Hepatotoxicity	Antioxidant effects
Panpatil et al., 2020 [342]	Wistar NIN (WNIN) rats	0, 50 and 100 ug/kg B.wt/day	Turmeric in diet 3% (wt/wt)	Liver and kidney	Decreasing DNA migration and genotoxicity
Apaydin et al., 2019 [343]	Albino rats	130 mg/kg B.wt/day	Curcumin—100 mg/kg B.wt/day Taurine—100 mg/kg B.wt/day	Cardiotoxicity	Antioxidant effects
Kalender et al., 2019 [344]	Wistar rats	130 mg/kg B.wt/day	Curcumin—100 mg/kg B.wt/day Taurine—100 mg/kg B.wt/day	Testicular toxicity	Antioxidant effects

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
