# Peer review of "Natural Products in Mitigation of Bisphenol A Toxicity: Future Therapeutic Use"

_molecules, 2022, doi:10.3390/molecules27175384_

Round 1

Reviewer 1 Report

The authors of the review titled: ‘Natural products in mitigation of bisphenol A toxicity: Future therapeutic use described the action of numerous biological materials and substances, which may be potentially useful in mitigation of toxic effects provoked by bisphenol A, which is one of the most important hazardous compound in the environment and human surrounding

Generally, I found this work interesting and valuable; however there are numerous comments that must be addressed by authors to improve the study.

Major comments

In my opinion section 3 should be divided to subsections. The first section should contain description of biological materials, such as plants, honey, propolis, etc. containing mixture of biological substances, and the second should focus on description of selected biological substances, like curcumin, resveratrol, etc.

Section 2 should be placed after Introduction and preceded by short description of the occurrence
of bisphenol A in surface water, soil, and the air with presentation of the concentrations of BPA
in these environmental compartments. It is also possible to combine this section with Introduction.

In numerous parts of the manuscript (particularly section 3), there is lack of information concerning the type of model (human, animal, plant) used in the experiments. For instance, line 383, 386, 415, 451, 464, 466, 488, 499, 510, 580, 582, 610, 612, 656, 679, 688, 699, 708, 720, etc.

In conclusions, the authors should suggest, which material(s), compound(s) seem to be the most promising in alleviating BPA toxicity. Does supplementation (with plant material/selected compounds) of workers occupationally (heavily) exposed to BPA should be particularly taken into consideration in order to decrease the risk of some diseases caused by this toxicant?

Minor comments  

Line 45, reference [5] does not address the occurrence of BPA in the environment. Please, provide
a few references on determination of BPA in surface water, soil, air. If authors decide to combine section with Introduction, the concentrations of BPA must also be added in this place.

Line 47-49, this sentence is not clear. From what material BPA leached into water?

Line 67, provide more references on BPA presence in the human body (including BPA concentrations)

Line 166, On the beginning of this section some references on the effect of BPA (prooxidative, genotoxic, proapoptotic) on blood cells may be mentioned. Usually, these changes are associated with immunotoxic potential of toxicants, like BPA.

Line 321, 325, correct to: P. integerrima

Line 387, 389, 392, correct to: A. spinosus

Line 566, I suggest to remove word ‘safe’ as there are not enough results showing no side-effects
of this material

Line 620, correct ‘material’ to ‘substance’

Line 672, correct to: ‘… BPA-induced apoptosis by …’

Line 688-691, correct sentence, I supposed that not only co-treatment of germ cells was studied.

Line 701, correct to: ‘GSH’

Author Response

Reviewer 1

The authors of the review titled: ‘Natural products in mitigation of bisphenol A toxicity: Future therapeutic use’ described the action of numerous biological materials and substances, which may be potentially useful in mitigation of toxic effects provoked by bisphenol A, which is one of the most important hazardous compound in the environment and human surrounding

Generally, I found this work interesting and valuable; however, there are numerous comments that must be addressed by authors to improve the study.

 Response: We thank the reviewer for the valuable comments. We have addressed all comments in the revised version.

Major comments

  1. In my opinion section 3 should be divided to subsections. The first section should contain description of biological materials, such as plants, honey, propolis, etc. containing mixture of biological substances, and the second should focus on description of selected biological substances, like curcumin, resveratrol, etc.

Response: As per suggestion, section 3 has been divided into two sections. The first section has plant extracts, and in the second section, the natural compounds are described.

  1. Section 2 should be placed after Introduction and preceded by short description of the occurrence of bisphenol A in surface water, soil, and the air with presentation of the concentrations of BPA
    in these environmental compartments. It is also possible to combine this section with Introduction.

Response: The section 2 was placed after introduction and a short description of the occurrence of bisphenol A in surface water, soil, and the air along with presentation of the concentrations of BPA was added (line 87 to line 134).

  1. In numerous parts of the manuscript (particularly section 3), there is lack of information concerning the type of model (human, animal, plant) used in the experiments. For instance, line 383, 386, 415, 451, 464, 466, 488, 499, 510, 580, 582, 610, 612, 656, 679, 688, 699, 708, 720, etc.

Response: The information regarding the type of animal model used in all the studies described in the table 1 and highlighted (line 318 to line 321).

  1. In conclusions, the authors should suggest, which material(s), compound(s) seem to be the most promising in alleviating BPA toxicity. Does supplementation (with plant material/selected compounds) of workers occupationally (heavily) exposed to BPA should be particularly taken into consideration in order to decrease the risk of some diseases caused by this toxicant?

Response: As per suggestion, the plant extracts/compounds that are most promising in alleviating BPA toxicity were mentioned in the conclusion (line 807 to line 811).

Minor comments  

Line 45, reference [5] does not address the occurrence of BPA in the environment. Please, provide a few references on determination of BPA in surface water, soil, air. If authors decide to combine section with Introduction, the concentrations of BPA must also be added in this place.

Response: Line 87 to line 106 has been added.

  1. Line 47-49, this sentence is not clear. From what material BPA leached into water?

Response: This statement has been corrected (line 46-line 49).

  1. Line 67, provide more references on BPA presence in the human body (including BPA concentrations)

Response: Line 69 to line 78 has been added.

  1. Line 166, On the beginning of this section some references on the effect of BPA (prooxidative, genotoxic, proapoptotic) on blood cells may be mentioned. Usually, these changes are associated with immunotoxic potential of toxicants, like BPA.

Response: Line 224 to line 235 has been added.

  1. Line 321, 325, correct to:  integerrima

Response: It is corrected

  1. Line 387, 389, 392, correct to:  spinosus

Response: It is corrected

  1. Line 566, I suggest to remove word ‘safe’ as there are not enough results showing no side-effects of this material

Response: It is corrected

  1. Line 620, correct ‘material’ to ‘substance’

Response: It is corrected

  1. Line 672, correct to: ‘… BPA-induced apoptosis by …’

Response: It is corrected

  1. Line 688-691, correct sentence, I supposed that not only co-treatment of germ cells was studied.

Response: The statement has been corrected and highlighted (576-580)

  1. Line 701, correct to: ‘GSH’

Response: It is corrected

Reviewer 2 Report

General comments

The manuscript entitled “Natural Products in Mitigation of Bisphenol A Toxicity: Future Therapeutic Use” gives a systematic overview of the effects of bisphenol A exposure on different body systems and about various natural compounds that are effective against bisphenol A-induced toxicity. It is a very interesting manuscript. This review clearly presented existing data about the toxicity of bisphenol A in reproductive, cardiovascular, endocrine, urinary, gastrointestinal, immune, respiratory, and nervous systems. The authors also gave an overview of the effects of different natural products such as resveratrol, luteolin, lycopene, quercetin, etc. that can be used to overcome the problems caused by BPA exposure.

Overall, the paper is well presented and organized and I would recommend it for publication.

Specific comments

Line 55: Please correct the information of daily tolerated intake of BPA. 4 g/kg/day is an error.

Author Response

Reviewer 2

General comments

The manuscript entitled “Natural Products in Mitigation of Bisphenol A Toxicity: Future Therapeutic Use” gives a systematic overview of the effects of bisphenol A exposure on different body systems and about various natural compounds that are effective against bisphenol A-induced toxicity. It is a very interesting manuscript. This review clearly presented existing data about the toxicity of bisphenol A in reproductive, cardiovascular, endocrine, urinary, gastrointestinal, immune, respiratory, and nervous systems. The authors also gave an overview of the effects of different natural products such as resveratrol, luteolin, lycopene, quercetin, etc. that can be used to overcome the problems caused by BPA exposure.

Overall, the paper is well presented and organized and I would recommend it for publication.

Specific comments

Line 55: Please correct the information of daily tolerated intake of BPA. 4 g/kg/day is an error.

Response: Thank you very much for your valuable comments. We corrected the daily tolerated intake of BPA. to 4 μg/kg/day.

Round 2

Reviewer 1 Report

Dear Editor,

Authors of the review titled: ‘Natural products in mitigation of bisphenol A toxicity: Future  therapeutic use significantly improved their manuscript. However, as I have indicated before, in numerous parts of the paper the information is (still) lacking on the kind of model organism used in the described experiments. Most of this information is now contained in table 1, but it must also be included in the body of the manuscript, and particularly
in section 3.

Sincerely Yours,

Jaromir Michałowicz

Author Response

Authors of the review titled: ‘Natural products in mitigation of bisphenol A toxicity: Future  therapeutic use” significantly improved their manuscript. However, as I have indicated before, in numerous parts of the paper the information is (still) lacking on the kind of model organism used in the described experiments. Most of this information is now contained in table 1, but it must also be included in the body of the manuscript, and particularly
in section 3.

Response: We are grateful to the reviewer and convey our thanks for the valuable comments. As per suggestion, we have included the information regarding animal model for described experiments (in section 3) of the manuscript and the relevant portions are highlighted in red color.